# Properties of Starve-Fed Extrusion on a Material Containing a VHMWPE Fraction

**DOI:** 10.3390/polym13060944

**Published:** 2021-03-19

**Authors:** Raffael Rathner, Davide Tranchida, Wolfgang Roland, Franz Ruemer, Klaus Buchmann, Philipp Amsüss, Georg Steinbichler

**Affiliations:** 1Institute of Polymer Extrusion and Compounding, Johannes Kepler University Linz, Altenberger Str. 69, 4040 Linz, Austria; wolfgang.roland@jku.at (W.R.); philipp.amsuess@jku.at (P.A.); Georg.Steinbichler@jku.at (G.S.); 2Borealis Polyolefine GmbH, Sankt-Peter-Straße 25, 4021 Linz, Austria; davide.tranchida@borealisgroup.com (D.T.); franz.ruemer@borealisgroup.com (F.R.); Klaus.Buchmann@borealisgroup.com (K.B.)

**Keywords:** single-screw extrusion, starve-fed extrusion, melting mechanism, VHMWPE

## Abstract

Single-screw extruders are usually operated with the screw fully filled (flood-fed mode) and not partially filled (starve-fed mode). These modes result in completely different processing characteristics, and although starve-fed mode has been shown to have significant advantages, such as improved mixing and melting performance, it is rarely used, and experimental studies are scarce. Here, we present extensive experimental research into starve-fed extrusion at feeding rates as low as 25%. We compared various operating parameters (e.g., residence time, pressure build-up, and melting performance) at various feeding rates and screw speeds. The results show a first insight into the performance of starve-fed extruders compared to flood-fed extruders. We explored starve-fed extrusion of a polyethylene material which contains a Very High Molecular Weight Polyethylene fraction (VHMWPE). VHMWPE offers several advantages in terms of mechanical properties, but its high viscosity renders common continuous melt processes, such as compression molding, ram extrusion and sintering, ineffective. This work shows that operating single-screw extruders in extreme starve-fed mode significantly increases residence time, melt temperature, and improves melting and that-in combination—this results in significant elongation of VHMWPE particles.

## 1. Introduction

Single-screw extruders are among the most important machinery in polymer processing. They are usually operated in flood-fed mode, where the screw beneath the hopper is fully filled and takes in as much material as possible. However, single-screw extruders can also be operated in starve-fed mode, where the screw is not fully filled. As its output depends on the feeding rate and not on the screw speed, a partially filled screw exhibits processing characteristics (e.g., axial pressure profile and melting) that differ completely from those of a screw operated in flood-fed mode [1]. The starve-fed mode is rarely used in single-screw extrusion, although significant benefits in terms of improved melting and mixing have been observed for twin-screw extruders operated in this mode. To date, only a few studies that concentrate on starve-fed single-screw extrusion have been published. The melting mechanism in starve-fed extrusion was discovered by various researchers. All these studies show a completely different melting mechanism compared to flood-fed mode. In starve-fed mode the melting mechanism consists of two stages. In the starve-fed areas conductive melting is the major mechanism while dispersed solid melting in the fully filled areas occurs [2,3,4,5,6]. Other groups were studying the improved mixing performance in compounding with starve-fed mode compared to flood-fed mode [7]. In addition, the processing advantages like limited die pressure fluctuations and a lower specific power consumption by starve-fed mode have been studied [8]. An optimization model for starve-fed single-screw extrusion showed that the optimized mode in a single-screw extruder in terms of specific energy consumption is starve-fed mode [9]. Recently Mazzanti showed the improvement in mechanical properties of rubber-toughened wood composites through starve-fed extrusion [10]. Compared to conventional polyolefins, very high molecular weight polyethylene (VHMWPE) offers several advantages in terms of mechanical properties, such as very high impact strength and crack resistance. However, its high viscosity makes pure VHMWPE unsuited to melting in common continuous processes such as compression molding, ram extrusion and sintering [11,12]. The processability of VHMWPE has been improved by decreasing the degree of chain entanglement [13], by applying atypical polymerization techniques (polymerization with catalysts with decreased number of active sites) [11,14,15,16,17] and by blending with lower-molecular-weight high-density polyethylene (HDPE) [17,18]. HDPE exhibits branching only to a very limited degree and consists of long chains that ensure its excellent mechanical properties [19,20]. Blending VHMWPE with polyethylene (PE) results in poor dispersion and thus in the formation of “white spots” due to the vast viscosity mismatch between the two components. White spots are VHMWPE particles that are too viscous for carbon black to penetrate, which results in phase separation [21]. Various (unsuccessful) attempts have been made to improve the morphology and homogeneity of VHMWPE/PE blends by melt blending and extrusion [22,23,24].

In the current work, we aimed to discover the effect of starve-fed extrusion on different processing parameters like melt temperature, residence time, degradation of the material, specific energy input and melting performance. Most notably, we were looking at the influence of starve-fed extrusion on PE blends which contain a VHMWPE fraction. Since blends containing a VHMWPE fraction are rather difficult to process we are showing the possibility to process these blends with starve-fed extrusion. To quantify the influence on starve-fed extrusion on the size and the shape of these particles we developed a program which can quantify both.

## 2. Materials and Methods

### 2.1. Materials

A black high-viscosity bimodal hexane copolymer polyethylene compound for HDPE pipes from Borealis with high density and an outstanding resistance to slow crack growth. The melt-flow rate (MFR) of this polymer measured according to ISO 1133 was 0.25 g/10 min (190 °C/5 kg). In the residence-time distribution measurements, we used this polymer in its natural (white) color and added 0.2 g of fluorescent dye (lunar yellow). In the melting-performance experiments, we used the same white polymer, to which we added 2% of carbon black masterbatch.

### 2.2. Extrusion Experiments

A single-screw extruder was used with a diameter of 45 mm and a length of 41 times the diameter that had a grooved-barrel feeding section. The extruder was equipped with seven pressure transducers at various axial positions, as listed in Table 1. All pressure sensors where from Gefran. P_1_–P_6_ had an upper limit of 2000 bar and P_7_ had an upper limit of 500 bar. A pressure valve at the screw tip was used to adjust the backpressure to specified values. A schematic representation of the extrusion line can be seen in Figure 1.

The temperature profile of the extruder (Table 2) was held constant across all experiments: The F0 zone extended to the end of the grooves of the barrel. Successive heating zones (Z1, Z2, Z3, Z4, Z5, and Z6) covered the whole barrel. The “tool” zone heated the die.

A commercially available barrier screw from KraussMaffei was used. The solids-conveying section, the melting or barrier section and the melt-conveying section were 500 mm, 650 mm and 450 mm long, respectively. The mixing section consisted of a double maddock and a distributive toothed mixing element and was 240 mm long. To control the amount of material fed to the single-screw extruder, we used a volumetric dosing system which feeds directly into the feed hopper of the extruder.

### 2.3. Experimental Overview

The experiments were performed at two different screw speeds and four different feeding rates. All parameters were evaluated for the operating points listed in Table 3.

For each operating point we analyzed:Output;Melt temperature;Specific energy input (SEI);Degradation after extrusion;Residence-time distribution;Melting performance;VHMWPE particle distribution;Shape distribution of the VHMWPE particles.

### 2.4. Measurement of Melt Temperature

The melt temperature of the extruded samples was measured inside the pressure valve by a temperature-measuring sword with a length of 7.60 mm and a width of 5.15 mm from Graeff GmbH.

### 2.5. Specific Energy Input

The drive power *P* of the extruder is calculated by:(1)$$P=M_{s}*2*\pi *N_{s},$$
where $M_{s}$ is the torque and $N_{s}$ the screw speed. The specific energy input (SEI), that is, the mechanical energy that must be provided by the screw torque to extrude 1 kg of polymer, is calculated by:(2)$$SEI=\frac{P}{ṁ}.$$

A lower specific energy input indicates that less mechanical energy is applied to the material.

### 2.6. Measurement of Degradation after Extrusion

An Anton Paar Plate-Plate Rheometer (MCR302) was employed using the frequency-sweep method from 628 rad/s down to 0.01 rad/s. Viscosity, storage module and loss module were determined for this range to investigate degradation effects. Measurements were carried out at 210 °C under nitrogen to avoid degradation during the rheology measurements using a parallel plate geometry with a diameter of 25 mm and a thickness of 0.8 mm. The pellet and all extruded samples were measured using the same method. Deviation of an extruded sample’s curve from that of the pellet is considered to signify degradation. In particular, the deviation at the zero-shear viscosity is a sign of degradation. The ratios of the viscosity of the pellet and the extruded samples at zero shear rate have been used to determine degradation.
(3)$$\eta^{*}=\frac{\eta_{Ext}}{\eta_{Pellet}}$$
where $\eta^{*}$ is the ratio of the viscosities at a shear rate of 0.01 s^−1^, $\eta_{Ext}$ is the viscosity of the extruded sample at a shear rate 0.01 s^−1^ and $\eta_{Pellet}$ is the viscosity of the pellet at a shear rate 0.01 s^−1^.

### 2.7. Measurement of Residence Time

To investigate the residence-time distribution in the extruder for a given setting, fluorescence spectroscopy was used [25]. An amount of 0.2 g of lunar yellow (fluorescent dye) was placed in the extruder entrance and determined its residence time. The fluorescence sensor measured the concentration of the fluorescent dye in the extruded material as a function of time. The residence time distribution function f(t) was calculated by Equation (4).
(4)$$f{\left(t\right)}_{\left(t\right)}=\frac{c\left(t\right)}{\int_{0}^{\infty }c\left(t\right)dt},$$
where *c*(*t*) is the fraction of molecules that have been in the extruder for time *t* or longer. The cumulative residence time distributions F(t) is given by Equation (5).
(5)$$F\left(t\right)=\overset{t}{\underset{0}{\int }}f\left(\tau \right)\;d\tau ,$$

### 2.8. Melting-Performance Analysis

To investigate the influence of various feeding rates on melting performance, screw-pulling tests were conducted. To this end, each operating point (i.e., a specified combination of screw speed and feeding rate) was configured and held for 30 min. Subsequently, the extruder was stopped abruptly and cooled down to room temperature. The screw was then pulled out of the extruder, and the solidified melt removed from the screw, cut into pieces, polished and finally scanned. The images were analyzed by transforming them into a black-and-white representation as shown in Figure 2, where the white and black parts, respectively, indicate the solid and the melt content. Samples were taken after each revolution.

### 2.9. Analysis of Sample Homogeneity 

To check whether extrusion increased the homogeneity of the materials, VHMWPE particles, which are suspended within—and are immiscible with—were tested in the polymer matrix. For this analysis, the extruded samples were cut by a Leica Rotation Microtome RM2265 into 12 µm-thick slices that were embedded in (Eukitt) resin between glass and cover slide. The resin was left to harden for 10 h, and then an Olympus SZX10 stereomicroscope equipped with a UC90 camera (Figure 3) was used to capture images at 10× and 20× magnification.

### 2.10. Analysis of Particle-Size Distribution

The images were analyzed by means of the Wavemetrics Igor Pro 8 software tool. In order to keep the analysis as objective as possible, code was written to repeatedly perform the same automated steps. The images were first converted from RGB to a gray-scale representation, and then the built-in procedure for bimodal distributions was used to determine a threshold from the intensity distribution. In cases in which this did not deliver satisfactory results, we used the built-in iterative method. Particles were identified in the binary image as ellipses, using the threshold and excluding particles smaller than 5-pixel^2^ or touching the edge of the image. Parameters recorded for each particle were area, circularity, and mean size of the ellipse axis.

### 2.11. Analysis of VHMWPE Particle-Shape Distribution

To determine the influence of starve-fed extrusion on the shape of VHMWPE particles, the circularity (roundness) (Ř) parameter was used, which is calculated by Equation (6).
(6)$$\text{Ř}=\frac{4*\pi *A}{P^{2}},$$
where A is the area and P the perimeter. Different shapes and corresponding circularity values can be seen in Figure 4.

## 3. Results and Discussion

### 3.1. Influence of Feeding Rate on Output

The influence of feeding rate on the output measured is illustrated in Figure 5. It can be seen that, at both screw speeds (50 rpm and 200 rpm), the output measured decreases with decreasing feeding rate. A 25% reduction in feeding rate results in a 25% reduction in output. The output at the feeding rate of 100% for 50 rpm was 48.7 kg/h and for 200 rpm was 197.6 kg/h. 

### 3.2. Influence of Feeding Rate on Melt Temperature

Figure 6 plots the melt temperature as a function of the feeding rate. A reduction in feeding rate has a great impact on the melt temperature: At a screw speed of 200 rpm, reducing the feeding rate from 100% to 25% increases the melt temperature from 220 to 291 °C. With decreasing feeding rate, the screw is no longer fully filled, and thus particles experience greater shear and remain longer within the extruder. Increases in shear and residence time result in a significant increase in melt temperature. With decreasing screw speed, the influence of feeding rate on melt temperature also decreases markedly.

### 3.3. Influence of Feeding Rate on Pressure Build-Up

Figure 7 illustrates the influence of the feeding rate on the axial pressure profile at a constant backpressure of 300 bar. At a feeding rate of 100% the extruder is completely filled, and at 25% feeding rate the first three pressure sensors are without pressure, which means that the screw is partially filled in the intake zone. With decreasing feeding rate, all other pressure sensors also show a drop in pressure. The results show that starve-fed extrusion results in a completely different pressure build-up.

### 3.4. Influence of Backpressure on Pressure Build-Up at Various Feeding Rates 

#### 3.4.1. Influence of Backpressure on Pressure Build-Up at 100% Feeding Rate 

Figure 8 plots the overall pressure build-up as a function of backpressure at a feeding rate of 100% for screw speeds of 50 rpm and 200 rpm; with rising backpressure, the pressure in all other zones increases, which is typical in extrusion.

#### 3.4.2. Influence of Backpressure on Pressure Build-Up at 25% Feeding Rate 

For comparison, Figure 9 illustrates pressure build-up versus backpressure for a feeding rate of 25% at screw speeds of 50 rpm and 200 rpm.

As in the previous case of 100% feeding rate, with increasing backpressure an overall increase in the pressure profile can be observed for both screw speeds investigated. With decreasing feeding rate, the pressure zone in the first few sections of the extruder depressurizes. An increase in backpressure increases the pressure in the extruder. For low feeding rates, the backpressure length increases with increasing backpressure. Due to the low feeding rate and the ensuing depressurization in the feeding section, the grooves of the extruder no longer affect the output-pressure behavior, which results in a transformation from grooved single-screw to smooth single-screw extruder behavior.

### 3.5. Influence of Feeding Rate on Specific Energy Input (SEI)

The relationship between feeding rate and specific energy input for screw speeds of 50 rpm and 200 rpm is shown in Figure 10.

The results clearly show that with decreasing feeding rate the specific energy input rises. A high SEI value indicates that more mechanical energy is put into the material, which may result in severe degradation of the material. In contrast to the melt-temperature, the specific energy input for the lower screw speed (50 rpm) also increases significantly with decreasing feeding rate. Hence, this indicates that for the lower screw speed the cooling unit of the single-screw extrusion machinery was able to compensate the increased mechanical energy input, but was unable to do so for the higher screw speed.

### 3.6. Analysis of Degradation by the Extrusion Process

The frequency-sweep method is a cheap and fast way of determining the degree by which the extruded material has degraded. The influence of screw speed and feeding rate on degradation is illustrated in Figure 11. The greater the deviation of a sample curve from that of the pellet (especially in the low-shear region), the greater the degradation of the material.

The ratios of the viscosities at a shear rate of 0.01 s^−1^
$\eta^{*}$ can be seen in Table 4.

While the degree of degradation is visible at a screw speed of 200 rpm and 25% feeding rate, it is practically non-existent at 50 rpm, independent of feeding rate and 200 rpm. Degradation can be caused by long residence time of the material inside the extruder (thermal degradation) or by high shear stresses or high shear rates (mechanical degradation).

### 3.7. Residence Time Distribution 

The residence time distribution of the different screw speeds with different feeding rates can be seen in Figure 12. 

The values of the mean residence time for screw speeds with different feeding rates can be seen in Table 5.

The results of the cumulative residence-time distribution measurements can be seen in Figure 13 and Table 6.

At lower screw speeds, the influence of the feeding rate is particularly pronounced. Figure 13 shows the cumulative residence time and Table 7 shows the cumulative residence times for various proportions of particles (%) observed. 

The average residence times for various screw speeds and feeding rates are listed in Table 8.

With decreasing feeding rate the width of the residence timer distribution function increases. An increase in the residence time distribution function results in enhanced elongation mixing. The mean residence time shows a significant increase when applying starve-fed mode for both screw speeds. The cumulative residence time increases significantly with both decreasing screw speed and decreasing feeding rate. Especially for very low feeding rates the residence time increases tremendously.

### 3.8. Analysis of Melting Performance

Figure 14 shows the melting behavior for various feeding rates.

A decrease in feeding rate results in accelerated melting of the material due to higher residence time and increased SEI to the material. For both screw speeds, we consider the optimal feeding rate in terms of melting performance to be 50%. For a feeding rate of 25% the melting performance decreases compared to a feeding rate of 50%. The pressure profile (c.f. Figure 6) reveals that at a feeding rate of 25% the screw is pressure-less in the intake section up to an axial position of approximately 20 D. Thus, a significant proportion of the screw is not used for melting.

### 3.9. Analysis of Sample Homogeneity 

In Figure 15 and Figure 16, the distributions of VHMWPE in the pellet and in samples extruded at screw speeds of 50 rpm and 200 rpm at various feeding rates can be seen. At very low screw speed (50 rpm) and low feeding rates (25% and 50%), the VHMWPE particles are markedly elongated, which has not been reported elsewhere. At high feeding rates, no significant influence on the shape of VHMWPE particles is evident. 

The different analyzed parameters for 50 rpm and 200 rpm with different feeding rates can be seen in Table 9 and Table 10.

The results clearly show that the extrusion process significantly improves the homogeneity of the samples in terms of number of particles, average size of particles and overall area covered. Further, reducing the feeding rate decreases particle size for screw speeds of 50 rpm. With decreasing feeding rate the overall area covered by particles is reduced.

### 3.10. Particle Shape

The circularity parameter was used to analyze the shape of the VHMWPE particles. A value of one indicates a perfect cycle, while lower values reflect elongation of the particle shape. All particles smaller than 45 µm^2^ were ignored, and the remaining ones were classified according to circularity. Table 11 and Table 12 list number of particles and mean and median values of circularity for samples extruded, respectively, at screw speeds of 50 rpm and 200 rpm at various feeding rates.

Figure 17 illustrates the shape distributions of VHMWPE particles in samples extruded at various screw speeds and feeding rates.

It can be seen that a reduction in feeding rate results in elongation of the VHMWPE particles. Since using different extrusion equipment, including high-performance screws (energy-transfer screw, wave screw) and various mixing elements, showed no influence on white-spot formation, we conclude that starve-feeding is a promising approach to dealing with this problem.

## 4. Conclusions

In summary, the results clearly show that the behavior of extremely starve-fed extruders differs completely from that of flood-fed extruders. A reduction in feeding rate gives rise to a significant increase in melt temperature due to higher shear and significantly longer residence time. In addition, the pressure build-up at constant backpressure is completely different for highly starve-fed extruders. While flood-fed grooved extruders show a constant backpressure build-up, starve-fed extruders exhibit no measurable pressure. This completely different pressure build-up behavior for the starve-fed mode was already observed [8]. The increases in residence time and melt temperature at lower feeding rates may result in better mixing and melting behavior of the extruder. The mechanism of the melting behavior and the different properties were already observed [2,3,4,5,6]. The use of a starve-fed single-screw extruder as compounder was already discovered by Isherwood [7]. Especially for highly viscous particles in the melt, such as the VHWMPE particles used in this work, a higher melt temperature results in lower viscosity and thus in particle deformation: at very low screw speed and feeding rate, the VHMWPE particles are significantly elongated. Since no satisfactory way of extruding PE blends that contain a VHMWPE fraction has been found to date, these results may unlock the extrusion of such materials. If significant elongation of the VHMWPE particles can be reached mechanical properties of the extruded parts can be significantly improved. An equal distribution results in equally distributed mechanical properties of the extruded parts [27,28]. The elongation of the VHMWPE particles through starve-fed extrusion has not been observed in literature before. Since starve-fed extrusion can be carried out rather easily, this is a suitable method for the extrusion of such materials in science and industries. Additionally, the starve-fed extrusion of these materials shows the decreasing number and size of VHMWPE particles. This means that starve-fed extrusion processing of these materials is a suitable way to increase the homogeneity of these materials.

## Figures and Tables

**Figure 1 polymers-13-00944-f001:**
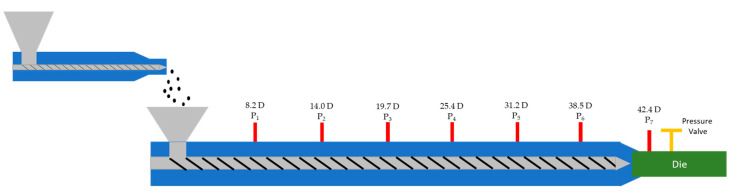
Schematic representation of extruder with feeder.

**Figure 2 polymers-13-00944-f002:**
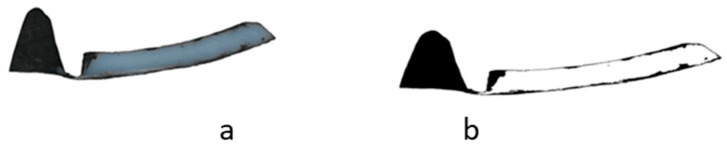
Scanned image of the melt cross section: (**a**) raw image, (**b**) after transformation to black and white.

**Figure 3 polymers-13-00944-f003:**
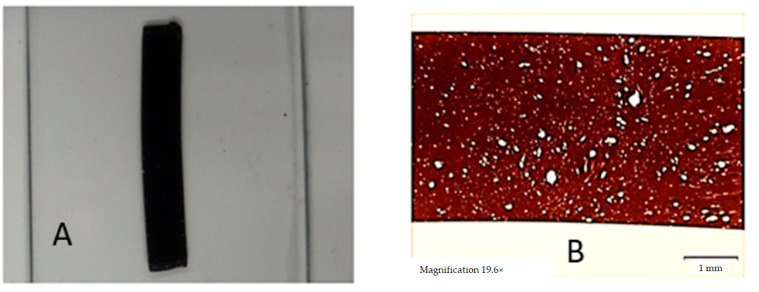
Representation of very high molecular weight polyethylene (VHMWPE) particle analysis. In (**A**) the representation of the sample preparation in (**B**) a microscope picture.

**Figure 4 polymers-13-00944-f004:**
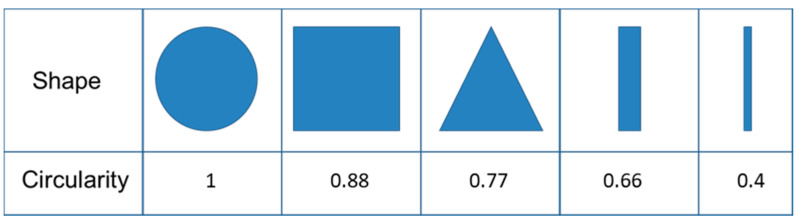
Shapes and their circularity values [26].

**Figure 5 polymers-13-00944-f005:**
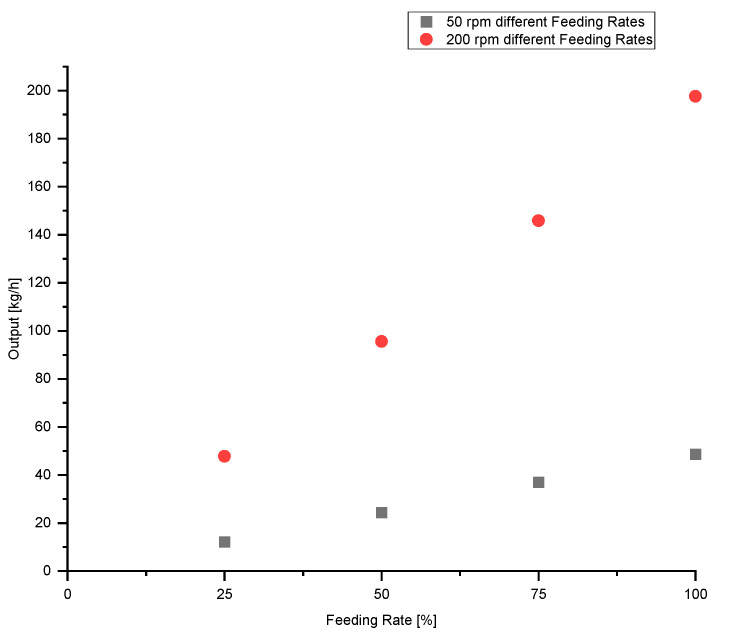
Influence of feeding rate on output at screw speeds of 50 rpm and 200 rpm.

**Figure 6 polymers-13-00944-f006:**
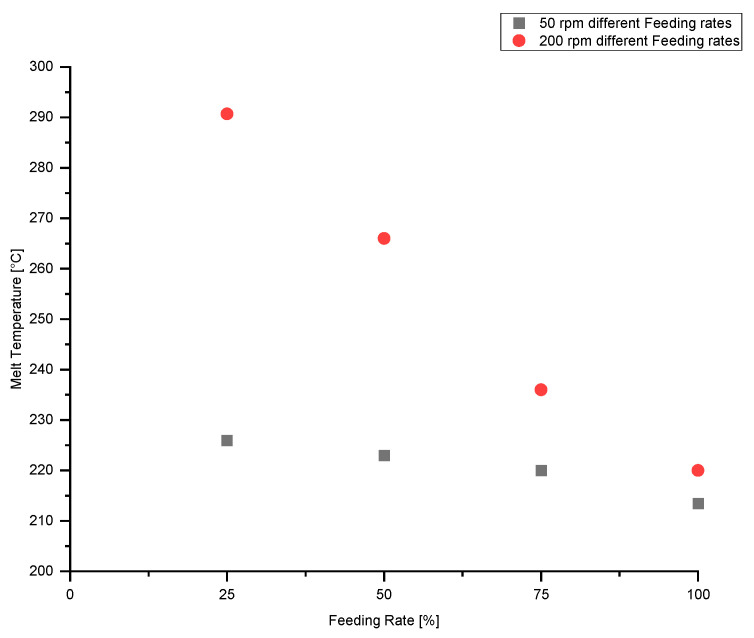
Influence of feeding rate on melt temperature at screw speeds of 50 rpm and 200 rpm.

**Figure 7 polymers-13-00944-f007:**
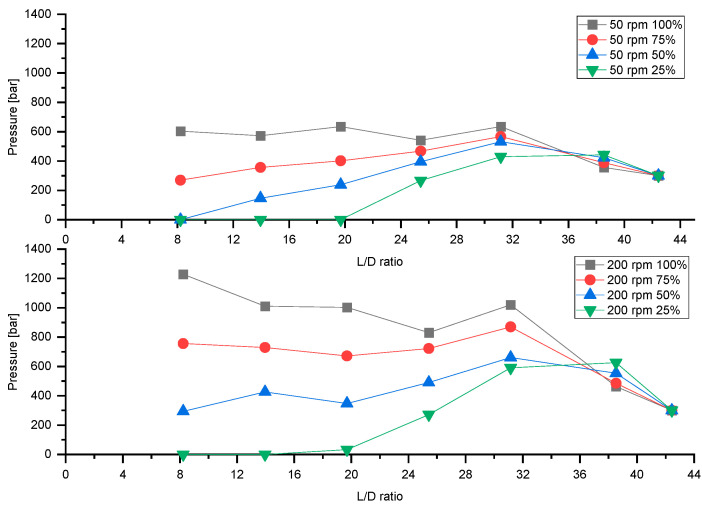
Influence of feeding rate on pressure build-up at screw speeds of 50 rpm and 200 rpm at a constant backpressure of 300 bar.

**Figure 8 polymers-13-00944-f008:**
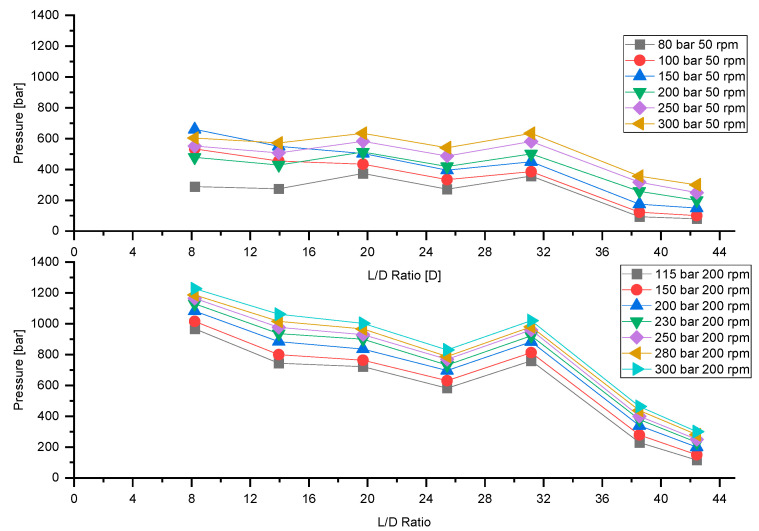
Influence of backpressure on pressure build-up at screw speeds 50 rpm and 200 rpm at a feeding rate of 100%.

**Figure 9 polymers-13-00944-f009:**
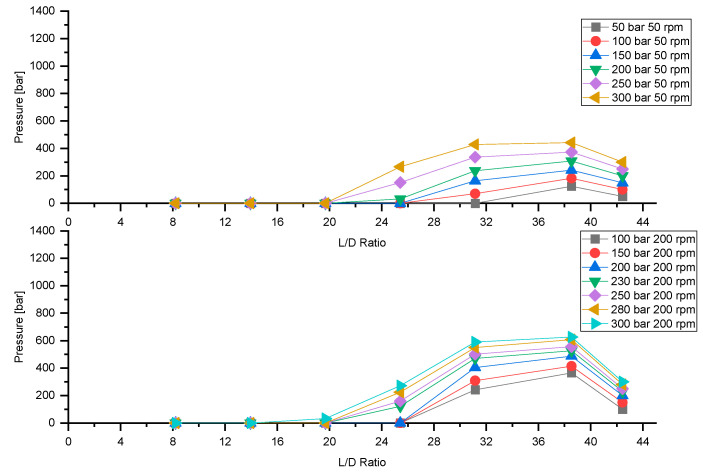
Influence of backpressure on pressure build-up at screw speeds 50 rpm and 200 rpm at a feeding rate of 25%.

**Figure 10 polymers-13-00944-f010:**
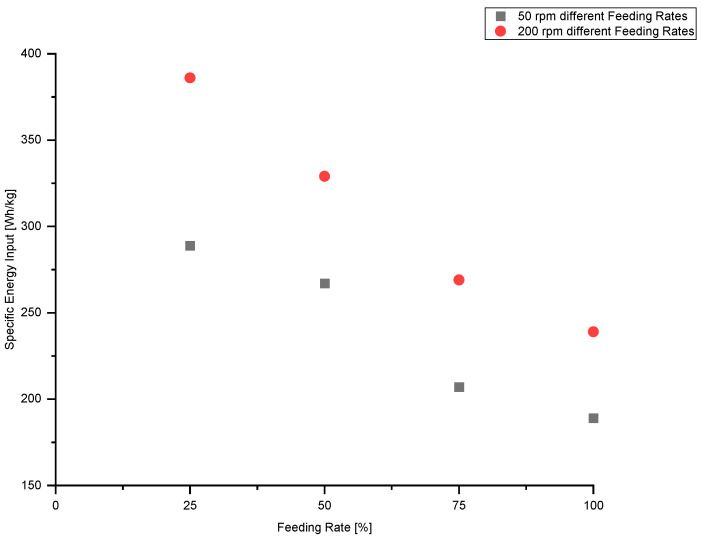
Influence of feeding rate on specific energy input (SEI) for two screw speeds.

**Figure 11 polymers-13-00944-f011:**
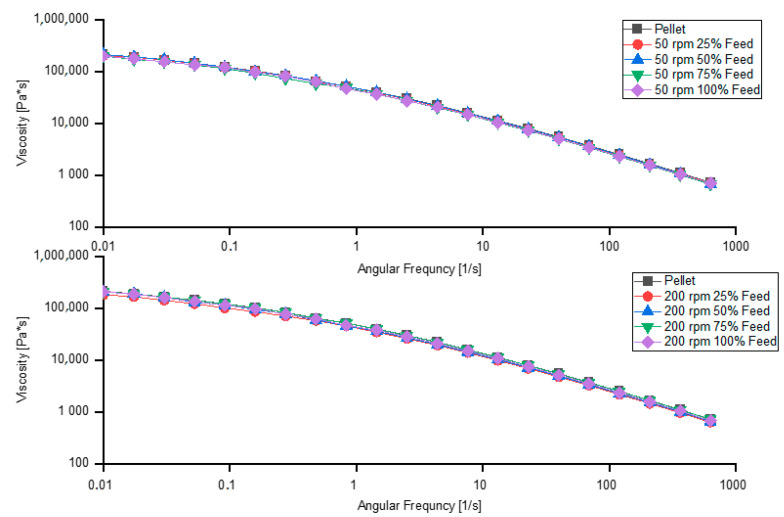
Degradation caused by extrusion process at screw speeds of 50 and 200 rpm and various feeding rates.

**Figure 12 polymers-13-00944-f012:**
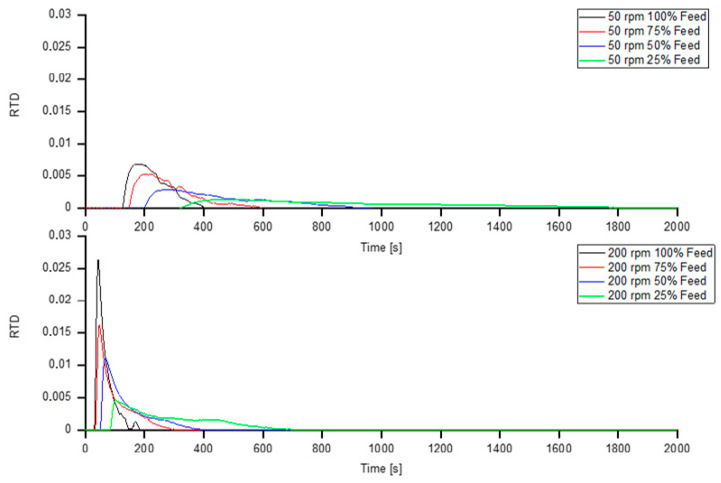
Residence-time distribution at screw speeds of 50 and 200 rpm for various feeding rates.

**Figure 13 polymers-13-00944-f013:**
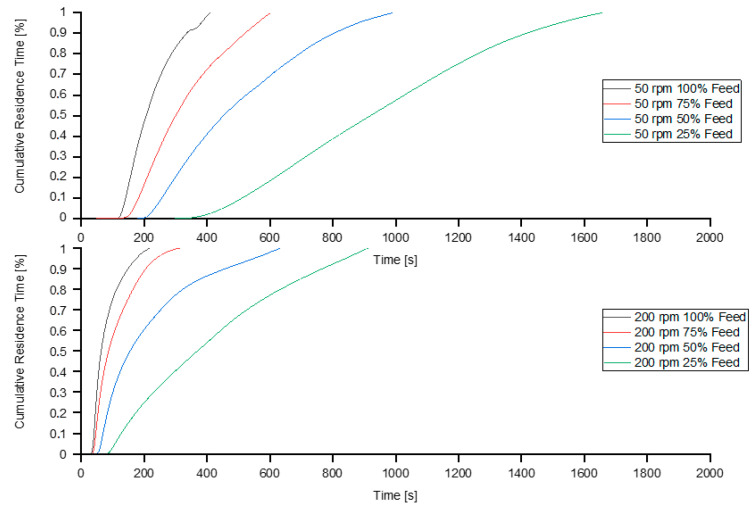
Cumulative Residence-time distribution at screw speeds of 50 and 200 rpm for various feeding rates.

**Figure 14 polymers-13-00944-f014:**
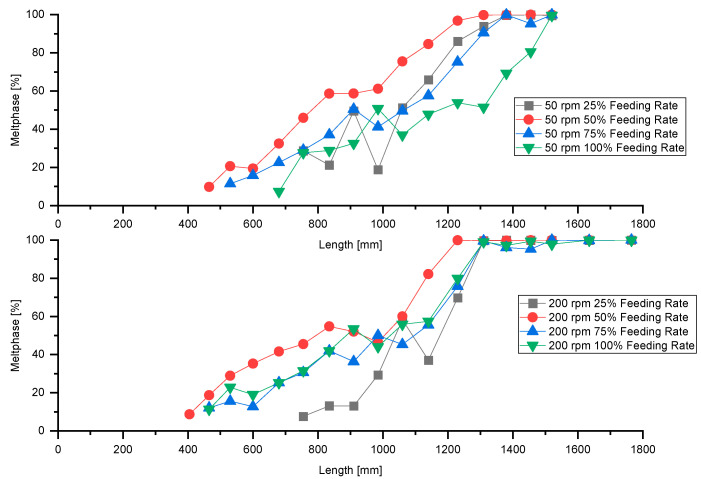
Melting performance for screw speeds of 50 and 200 rpm at various feeding rates.

**Figure 15 polymers-13-00944-f015:**
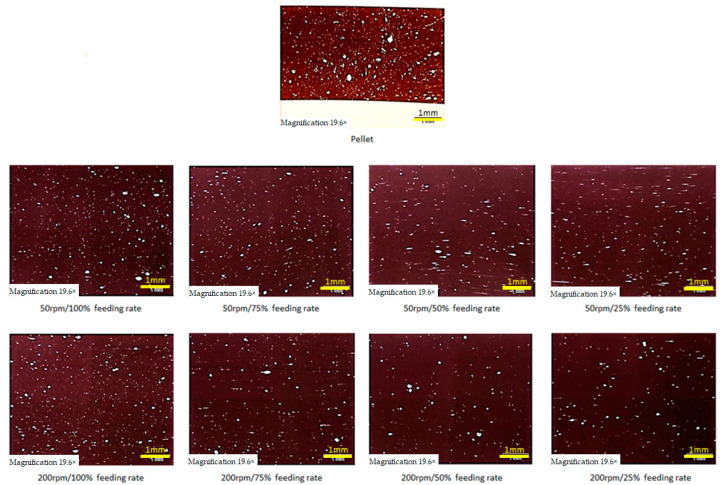
VHMWPE particles in pellet and samples extruded at screw speeds of 50 rpm and 200 rpm at various feeding rates.

**Figure 16 polymers-13-00944-f016:**
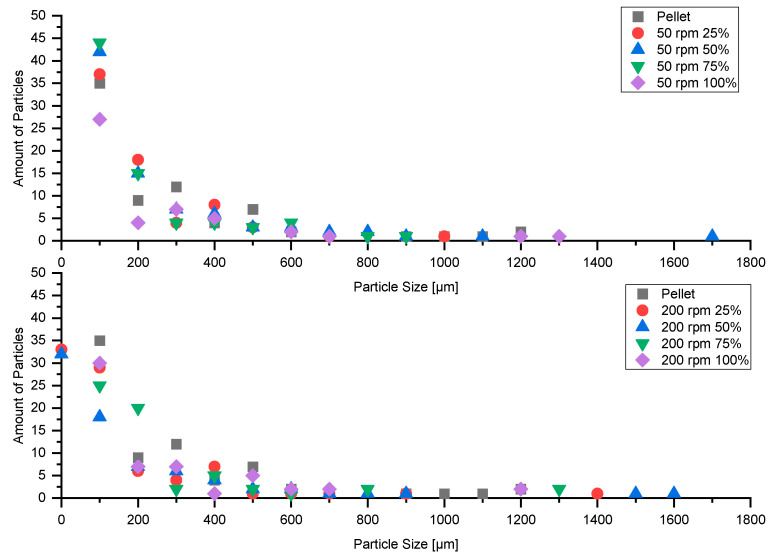
VHMWPE particles in pellet and samples extruded at screw speeds of 50 rpm and 200 rpm at various feeding rates.

**Figure 17 polymers-13-00944-f017:**
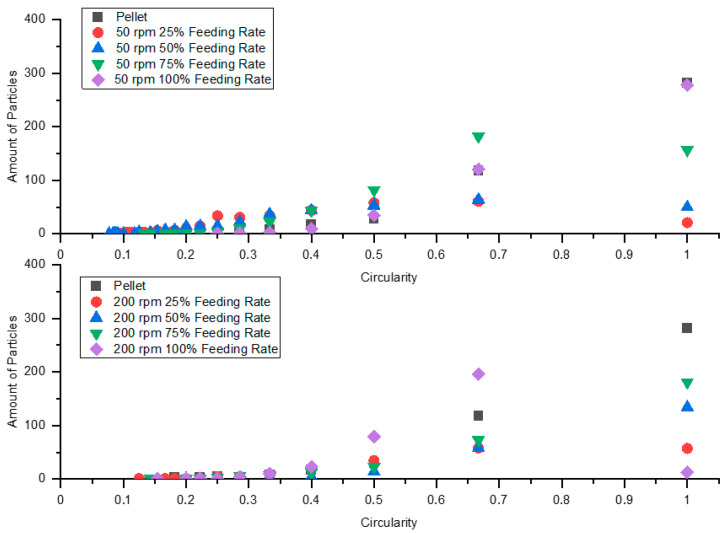
VHMWPE particle-shape distribution in samples extruded at screw speeds of 50 rpm and 200 rpm at various feeding rates.

**Table 1 polymers-13-00944-t001:** Positions of the pressure sensors along the grooved barrel given in multiples of the extruder diameter (D).

Pressure Sensor	Position (D)
P_1_	8.2
P_2_	14.0
P_3_	19.7
P_4_	25.4
P_5_	31.2
P_6_	38.5
P_7_ (backpressure)	42.5

**Table 2 polymers-13-00944-t002:** Temperature zones of the extruder.

Zone	Temperature [°C]
F_0_	40
Z_1_	220
Z_2_	215
Z_3_	210
Z_4_	200
Z_5_	200
Z_6_	200
Tool	200

**Table 3 polymers-13-00944-t003:** Operating points used in the experiments.

Screw Speed (rpm)	Feeding Rate (%)
50	25
50	50
50	75
50	100
200	25
200	50
200	75
200	100

**Table 4 polymers-13-00944-t004:** Ratios of the viscosities at a shear rate of 0.01 s^−1^ ($\eta^{*}$ ).

Screw Speed (rpm)	Feeding Rate	$\eta^{*}$
50	25	0.95
50	50	0.99
50	75	0.93
50	100	0.93
200	25	0.85
200	50	0.96
200	75	0.99
200	100	0.98

**Table 5 polymers-13-00944-t005:** Mean residence time for screw speeds of 50 rpm and 200 rpm and various feeding rates.

Feeding Rate	50 rpm	200 rpm
25%	877 s	189 s
50%	450 s	141 s
75%	281 s	102 s
100%	224 s	67 s

**Table 6 polymers-13-00944-t006:** Influence of screw speed on cumulative residence-time distribution at screw speeds of 50 rpm and 200 rpm and 100% feeding rate.

Percentage of Particles	50 rpm, 100% Feeding Rate	200 rpm, 100% Feeding Rate
10%	141 s	41 s
30%	171 s	50.5 s
50%	207 s	65.5 s
75%	269 s	105.5 s
100%	412,5 s	216 s

**Table 7 polymers-13-00944-t007:** Influence of screw speed on cumulative residence-time distribution at screw speeds of 50 rpm and 200 rpm and 25% feeding rate.

Percentage of Particles	50 rpm, 100% Feeding Rate	200 rpm, 100% Feeding Rate
10%	512 s	127.5 s
30%	713 s	231 s
50%	920 s	369.5 s
75%	1197 s	577 s
100%	1658 s	912.5 s

**Table 8 polymers-13-00944-t008:** Average Residence Time for screw speeds of 50 rpm and 200 rpm and various feeding rates.

Feeding Rate	50 rpm	200 rpm
25%	920 s	369.5 s
50%	452.5 s	155.5 s
75%	300 s	87 s
100%	207 s	65.5 s

**Table 9 polymers-13-00944-t009:** Parameters of the particles observed in samples extruded at a screw speed of 50 rpm at various feeding rates.

Sample	Number of Particles	Mean Particle Area (µm^2^)	Median Particle Size (µm)	Area Covered by All Particles (µm^2^)
Pellet	353	686.6	361.0	73,170
25% Feeding rate	141	532.5	352.0	22,665
50% Feeding rate	148	678.1	390.5	30,297
75% Feeding rate	158	575.2	355.5	27,434
100% Feeding rate	129	686.5	330.0	26,738

**Table 10 polymers-13-00944-t010:** Parameters of the particles observed in samples extruded at a screw speed of 200 rpm at various feeding rates.

Sample	Number of Particles	Mean Particle Area (µm^2^)	Median Particle Size (µm)	Area Covered by All Particles (µm^2^)
Pellet	353	686.6	361.0	73,170
25% Feeding rate	84	678.4	426.5	17,202
50% Feeding rate	76	807.7	449.5	18,533
75% Feeding rate	120	659.7	332.5	23,899
100% Feeding rate	162	704.1	356.0	34,433

**Table 11 polymers-13-00944-t011:** Parameters of the particles observed in samples extruded at a screw speed of 200 rpm at various feeding rates.

Sample	Number of Particles	Mean Value of Circularity	Median Value of Circularity
Pellet	353	0.6142	0.66
25% Feeding rate	141	0.2753	0.25
50% Feeding rate	148	0.3219	0.27
75% Feeding rate	158	0.5737	0.59
100% Feeding rate	129	0.6003	0.612

**Table 12 polymers-13-00944-t012:** Parameters of the particle-shape distribution in samples extruded at a screw speed of 200 rpm at various feeding rates.

Sample	Number of Particles	Mean Value of Circularity	Median Value of Circularity
Pellet	353	0.6142	0.66
25% Feeding rate	84	0.5326	0.55
50% Feeding rate	76	0.6227	0.65
75% Feeding rate	120	0.5934	0.63
100% Feeding rate	162	0.5863	0.61

## Data Availability

The data presented in this study are available on request from the corresponding author.

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
