# Peer review of "Properties of Starve-Fed Extrusion on a Material Containing a VHMWPE Fraction"

_polymers, 2021, doi:10.3390/polym13060944_

Round 1

Reviewer 1 Report

This work is interesting and in the scope of this journal. Nevertheless, it is necessary some modifications before its publication.

1) The discussion of previous works in this field must be included in the introduction section.

2) Avoid the use of personal sentences. Substitute them with passive sentences.

3) Line 66: "added a fluorescent dye" How much?

4) A schematic figure of the extrusion could facilitate its compresion.

5) What is the geometry of the nozzle in the extrusion?

6) The maximum feeding rate is necessary to evaluate what rate is a 25% of it.

7) Line 111: diameter?

8) Figure 14: In this figure it is impossible to observe the scale bar.  In addition, number them to identified them.

9) Discussion section is now a simple conclusion. In this section authors must discuss the obtained results and compare them with other works. In addition they could postulate hypothesis about the behavior observed in the results, comparing the different srew speed and feeding rate evaluated.

10) More recent references could be incorporated.

Author Response

1) The discussion of previous works in this field must be included in the introduction section.
I added new references from line 35

2) Avoid the use of personal sentences. Substitute them with passive sentences.
I changed in line 64,72,85,128,129,138,142,150,172,352 the language to passive

3) Line 66: "added a fluorescent dye" How much?
I added in line 68 0,2g

4) A schematic figure of the extrusion could facilitate its compresion.
I have added a schematic picture which now figure 1

5) What is the geometry of the nozzle in the extrusion?
We used a pressure valve as die where could adjust the die pressure manually (line 77)

6) The maximum feeding rate is necessary to evaluate what rate is a 25% of it.
You see the maximum Feeding rate in figure 4.  I also added a line in 196.

7) Line 111: diameter?
I added in line 119 the plate geomtry

8) Figure 14: In this figure it is impossible to observe the scale bar.  In addition, number them to identified them.
I added a new scale bar.

9) Discussion section is now a simple conclusion. In this section authors must discuss the obtained results and compare them with other works. In addition they could postulate hypothesis about the behavior observed in the results, comparing the different srew speed and feeding rate evaluated.

I added lines to the conclusion and I am discussing literature with our results 

10) More recent references could be incorporated.
I added more references which are in relation to our work

Reviewer 2 Report

The paper “Properties of Starve-fed extrusion on a material containing a VHMWPE fraction” is very interesting, novel, centered on the scope of the journal. The paper is suitable to publish buti it need some modification.

I know that the literature about starve-fed mode is very few, for which i suggest to the authors to add this new paper where a real benefit is observed in the use of this extrusion method:

Correlation between mechanical properties and processing conditions in rubber-toughened wood polymer composites Mazzanti, V., Malagutti, L., Santoni, A., Sbardella F., Calzolari A., Sarasini, F., Mollica, F. Polymers, 2020, 12(5), 278

Pag 2 line 62-68 The commercial name of the polymer and the all additives need to be added.

Pag 2 line 72 The trade name and the full scale values of each pressure transducers need to be added. Describe if, in addition to the pressure measurement, the transducers also acquire the temperature of the fluid

Pag 3 line 101 the authors need to indicate what type of instrument for measuring the temperature was used and its full scale

Since the results are numerous I ask the authors to discuss them within the results section and rename it “results and discussions” section. I also ask to expand them with respect to the small section present now. In addition, the paper must be concluded with a section called “conclusions”. Here the authors will describe the main conclusions of their research

Author Response

The paper “Properties of Starve-fed extrusion on a material containing a VHMWPE fraction” is very interesting, novel, centered on the scope of the journal. The paper is suitable to publish buti it need some modification.

I know that the literature about starve-fed mode is very few, for which i suggest to the authors to add this new paper where a real benefit is observed in the use of this extrusion method:

Correlation between mechanical properties and processing conditions in rubber-toughened wood polymer composites Mazzanti, V., Malagutti, L., Santoni, A., Sbardella F., Calzolari A., Sarasini, F., Mollica, F. Polymers, 2020, 12(5), 278

I added the work and some other literature to our work with some other work from line 35.

Pag 2 line 62-68 The commercial name of the polymer and the all additives need to be added.

It is a non commercially available research grade from borealis which produced in the full scale plant and is representative of a common PE100 grade. I am not allowed to give you more information according to Borealis.

Pag 2 line 72 The trade name and the full scale values of each pressure transducers need to be added. Describe if, in addition to the pressure measurement, the transducers also acquire the temperature of the fluid.

I added this in line 74-75. The transducers are pure pressure sensors

Pag 3 line 101 the authors need to indicate what type of instrument for measuring the temperature was used and its full scale. I added it in line 118

Since the results are numerous I ask the authors to discuss them within the results section and rename it “results and discussions” section. I also ask to expand them with respect to the small section present now. In addition, the paper must be concluded with a section called “conclusions”. Here the authors will describe the main conclusions of their research

I added lines to the conclusion and I am discussing literature with our results 

Round 2

Reviewer 1 Report

The authors have made all the reviewer's comments. So, I recommend its publication in the present form.